# Measurement System for Unsupervised Standardized Assessment of Timed “Up & Go” and Five Times Sit to Stand Test in the Community—A Validity Study

**DOI:** 10.3390/s20102824

**Published:** 2020-05-15

**Authors:** Sebastian Fudickar, Sandra Hellmers, Sandra Lau, Rebecca Diekmann, Jürgen M. Bauer, Andreas Hein

**Affiliations:** 1Assistance Systems and Medical Device Technology, Carl von Ossietzky University Oldenburg, 26129 Oldenburg, Germany; Sandra.Hellmers@uol.de (S.H.); Rebecca.Diekmann@uol.de (R.D.); andreas.hein@uol.de (A.H.); 2Center for Geriatric Medicine, Agaplesion Bethanien Krankenhaus Heidelberg, University Heidelberg, 69117 Heidelberg, Germany; sandra.lau@uni-heidelberg.de (S.L.); Juergen.Bauer@bethanien-heidelberg.de (J.M.B.)

**Keywords:** Timed “Up & Go” Test, TUG, Five Times Sit-to-Stand Test, 5xSST, machine learning, assessment, unsupervised, functional, system usability, validity analysis, evaluation, inertial measurement units, technology

## Abstract

Comprehensive and repetitive assessments are needed to detect physical changes in an older population to prevent functional decline at the earliest possible stage and to initiate preventive interventions. Established instruments like the Timed “Up & Go” (TUG) Test and the Sit-to-Stand Test (SST) require a trained person (e.g., physiotherapist) to assess physical performance. More often, these tests are only applied to a selected group of persons already functionally impaired and not to those who are at potential risk of functional decline. The article introduces the Unsupervised Screening System (USS) for unsupervised self-assessments by older adults and evaluates its validity for the TUG and SST. The USS included ambient and wearable movement sensors to measure the user’s test performance. Sensor datasets of the USS’s light barriers and Inertial Measurement Units (IMU) were analyzed for 91 users aged 73 to 89 years compared to conventional stopwatch measurement. A significant correlation coefficient of 0.89 for the TUG test and of 0.73 for the SST were confirmed among USS’s light barriers. Correspondingly, for the inertial data-based measures, a high and significant correlation of 0.78 for the TUG test and of 0.87 for SST were also found. The USS was a validated and reliable tool to assess TUG and SST.

## 1. Introduction

Muscle weakness and inactivity are strong predictors for developing functional disabilities [1,2], which might lead to restrictions in the Activities of Daily Living (ADL) [3]. Limitations in the ability to handle ADLs also lead to higher mortality [4]. Preserving or even initiating physical activity and exercises in older adults can stop, reduce, or reverse the loss of muscle mass [5] and increase physical capacity. Additionally, being physically active also has a positive impact on the quality of life [6], cognitive status [7,8], and frailty [9,10]. Although there is an increasing number of interventions designed to promote physical activity in older persons, the adherence to participating in exercise programs regularly is a common challenge [11]. To initiate appropriate interventions to avoid further physical declination, loss of mobility, and mortality, comprehensive measurements are required to detect functional decline at the earliest possible stage [12,13]. This requirement is especially true for the assessment of mobility, muscle power, and balance, as these functional parameters are closely linked to the ability to live independently [14,15].

Established comprehensive geriatric assessments of these functional parameters like the Timed “Up & Go” (TUG) Test [16] and the Five Times Sit To Stand Test (5SST) (also described as the Five Times Chair Rise Test) [17] are sensitive predictors for the development of disability [18] and recurrent falling [19], and therefore, these tests are well suited for this purpose. Consequently, the combination of TUG and 5SST could be a sensitive screening assessment to detect all three parameters associated with functional decline, even in a not yet acute stage in older adults.

To assure the early detection of functional decline, continuous measurements (e.g., monthly) may be preferred, but can become challenging from an organizational viewpoint due to time- and cost-related reasons. Regular screenings by medical practitioners (e.g., yearly screenings for fall risks as suggested by the American Geriatrics Society [20]) are often challenging to conduct during a time-framed examination. Completing the TUG and the 5SST only takes a few minutes, but according to test protocols, a trained supervisor is still required. Integrating frequent, supervised functional assessments to detect functional changes into routine care seems inapplicable. Therefore, an ongoing individual self-assessment might be an innovative and more effective approach to detect early changes in functional performance.

Unsupervised assessments via technical screening systems may provide a more suitable path with great potential. In theory, the optimal screening method would be a continuous data collection, e.g., by wearable devices. While various systems were proposed for the extraction of stair climb power [21] and mobility in general [22,23], the monitored biomechanical parameters are most meaningful within a defined context. However, monitoring systems for home-use are susceptible to unrecognized contextual variations [22]. Therefore, a frequently conducted assessment in a standardized setting (e.g., within a medical screening) is more suitable and assures higher intra-test reliability.

Applying automated assessments via technical screening systems such as the ambient TUG (aTUG) chair has been confirmed to be sensitive to detect functional decline [24], and it is also assumed to increase the intra-test reliability among consecutive measures. While still requiring the presence of a supervisor for advising on the procedure and supervision of correct execution, the aTUG includes already automated measuring functionality. The aTUG measures the TUG via an infrared Light Barrier (LB), four Force Sensors (FS), and a Laser Range-Scanner (LRS). Via these sensors, the aTUG can automatically measure the total time of the TUG and of all sub-tasks of the instrumented TUG (iTUG), as proposed by Botolfson et al., (2008) [25]. Similarly, the recently proposed Short Physical Performance Battery (SPPB) kiosk [26] is intended for supervised assessments of the SPPB with its three components (namely gait speed, the 5SST, and standing balance) and intends to enhance inter-tester reliability in executing the SPPB protocol. With semi-automatic post-processing, SPPB kiosk’s validity to estimate these SPPB-components was shown. Similarly, the applicability of Inertial Measurement Unit (IMU)-based wearable sensors to measure TUG and the 5SST performance automatically by measuring the acceleration and gyroscopic turning rate was confirmed [17,27].

Thus, a system for repeated and fully unsupervised reliable TUG and 5SST measurements by older individuals in community settings or as part of therapeutic rehabilitation is still needed. While most of these corresponding technical screening tools are well suited for guided assessments (e.g., in community settings), they still require the (tele-)presence of physicians or therapists, since correctly performing the movements cannot be assured, but is mandatory to ensure the comparability of the results.

Consequently, the validity of monitoring systems for the unsupervised frequent self-screening of the functional status (e.g., via TUG and 5SST) by older adults that do not require the presence of an expert has yet to be confirmed.

The proposed Unsupervised Screening System (USS) aims to support frequent (e.g., monthly) self-assessments in community-dwelling older adults via the time-based metrics of the TUG and 5SST in an unsupervised and standardized manner. The system is envisioned to empower older adults to screen their current functional status frequently and identify increased risk for functional decline at an early stage. As a consequence, using the USS might strengthen the users’ awareness to start interventions at an earlier stage following the chance of full recovery, which can improve self-determined living and potentially lower overall healthcare costs. The USS is intended for installation in public spaces such as community centers, fitness centers, and medical environments, where users have free access. The current system represents the initial prototype, and its form factor can be significantly reduced in later versions.

Correspondingly, the article at hand introduces the screening system for unsupervised functional assessments (USS) and evaluates the systems’ technical validity with the usually timed measurement done in the TUG and 5SST assessments by a cohort study of 91 older adults compared to manual stopwatch measures as a point of reference. With this article focusing on the technical validity, the user-experience of USS’s user-interfaces will be analyzed in another article. The USS integrates various existing components such as IMU-based activity recognition and the aTUG, which have been initially designed to support supervised assessments, and combines them with further sensors, new user-interfaces, and additional processing steps to ensure a clinically meaningful sensitivity in unsupervised assessments.

## 2. Materials and Methods

To evaluate the validity of the Unsupervised Screening System (USS) (introduced in Section 2.1), the corresponding machine-learning models for movement classification (described in Section 2.2), the study design (Section 2.3), and the post-processing methods (Section 2.4) are applied.

### 2.1. The Unsupervised Sensor System

The Unsupervised Sensor System (USS) (shown in Figure 1) guided users through the unsupervised performance of the TUG and 5SST and meanwhile screened their performance (The usage of the USS is shown in Appendix A). Within the USS, the user initially registered/logged in at the USS via the RFID scanner (a). Thereby, users could authenticate themselves by holding a small RFID dongle, e.g., attached to their key-ring, on the reader and did not need to remember a username and password. Afterward, they took a seat in the USS and followed the instructions shown on the touchscreen monitor (b). For intuitive user interaction, the USS used RFID-based user authentication with a touch-based display. An explanatory audio stream and a static visualization combined to present instructions on how to prepare for screenings and how to conduct the tests (as shown in Figure 2). Users could also choose to watch detailed video-based instructions. These video tutorials were also displayed in cases where a user did not complete the test correctly. Upon successful completion of the tests, the users were reminded to return the sensor belt and were automatically logged out by the system.

The USS (shown in Figure 1) monitored user movements via the following sensors: the aTUG chair [24] (c), a hip-worn inertial sensor (d), light barriers (e, c), and the Intel^®^ RealSense™ D435 depth image cameras (f).

The hip-worn inertial sensor was integrated into a sensor-belt and included a triaxial accelerometer, gyroscope, magnetometer, and a one-dimensional barometer (see Table 1) and should be aligned between the L3 and L5 lumbar vertebral body. The inertial sensor’s orientation is illustrated in Figure 3. The correct placement and orientation of the sensor belt was indicated by a yellow stripe on the belt and addressed within USS’s instructional videos. The inertial sensor was calibrated, and the data were correspondingly pre-processed. No orientation correction was conducted.

As reliable ambient sensors, light barriers (LBs) were applied in the aTUG chair and at a 3 m walking distance (turning distance) to measure the TUG, iTUG, and 5SST repetitions.

PCs, running Ubuntu 16.04 and Intel^®^ RealSense™ SDK 2.0 as librealsense (Build 2.11.1), were used as the computing backbone. The main application was built via QT 5.10.1, and a MySQL 5.7.24 instance was used as a local database. These datasets included the data of the sensor belt, depth images (with a resolution of 640 × 480 px, 30 Hz, and a depth of Z16) stored as bag files, the LRS scans, and the recordings of the LBs, pressure sensors, and the device interactions (such as log-in times and reaction/interaction delays).

The following termination criteria were used to detect incomplete or incorrect performances of the assessments (following the test procedures in the second paragraph of Section 1). Besides the detection of the re-connection of the sensor belt during the assessments, the considered termination criteria were detected based on the primary LB (a 61mcu.com E18-D80NK-N, connected via the Arduino) and the LB at a 3 m distance. For the assessments, the termination criteria leading to the detection of incorrect test performance were:(TUG) Delayed beginning of assessment (standing up) over at least 3 s, as detected by the opening time of the primary LB.(TUG) Not crossing the 3 m distance line completely, as indicated by twice consecutive activation of the LB at a 3 m distance.(TUG) Extended assessment duration of the TUG over 30 s, which might result from not leaning backward on the TUG chair, recognized by the primary LB not being triggered again after crossing the LB at a 3 m distance.(5SST) Less than five times repetition of the sit-to-stand sequence as measured via the primary LB.(5SST) Surpassing a maximal assessment duration of 40 s.

In these cases, the participants had to repeat either the TUG and/or 5SST after having watched an extended introduction video explaining how to conduct the assessment. This extended version covered potential errors in greater detail for the correct test performance.

### 2.2. Machine Learning Model for Movement Classification

With the LB-based evaluation being straight forward, Figure 4 shows the resulting classification and measurement workflow for the inertial raw data measures.

The Machine-Learning (ML) Model was trained on an annotated dataset of a previous study covering movements of older adults over 70 years while conducting functional assessments in a supervised setting [28]. The data of the training set were recorded with the same IMU. The training set covered recorded activities of 184 subjects (145 older and 39 younger adults). Among the considered older adults, some of the subjects of the given study most likely were included. Within an evolutionary optimization process, models for various ML approaches were optimized, resulting in the most suitable model per activity group (static, dynamic, transition, and general state) being identified. The following input features (each per axis of the accelerometer, the gyroscope, and magnetometer) were considered:Root Mean Square (RMS)MeanSignal Energy (SE)Autocorrelation (AC)Correlation (C)Signal Magnitude Area (SMA)Standard Deviation (SD)Pitch

A detailed description of each feature and its calculation can be found in [29].

Additionally, a low-pass filter (cut-off frequency: $f_{c}$ = 4.5 Hz) was used for noise reduction. While the general state classification used boosted decision trees, static, dynamic, and transition states were best classified via multilayer perceptrons (as shown in Table 2). Further details were discussed in [27].

The ML models were trained in advance (as reported in [27]) on a separate dataset with clinically supervised movements that covered a similar age-related group characteristics and thus were applied to classify the movements in this unsupervised setting. The trained ML models were used in the activity classifier to classify the IMU raw data and generate activity labels (see Figure 4).

To detect sequences of the considered assessments automatically, the classified motion labels (as the output of the trained classifiers) were processed regarding valid sequential activities via the following rule-based approach.

Following [17,27], the sequence of activities and transitions, as summarized in Table 3, were valid and covered variations of performing the TUG and 5SST. For the TUG, additional sequence permutations were included to cover common classification errors unrelated to subjects’ performance errors. All other combinations of motion labels during test sequences represented erroneous performances. For the valid test sequences, the performance analyzer algorithm determined and reported the assessment duration as the summed durations of the considered motion labels for the test sequence.

### 2.3. Study Design

To evaluate the validity of the time-based LB and IMU measures within the USS, the TUMALstudy was conducted. Within the TUMAL study, participants of the previous VERSAstudy [28] were considered based on their expressed interest to continue participation. Potentially interested participants were called to confirm in a phone interview that they fulfilled the inclusion criteria. The following inclusion criteria were applied:age ≥ 70 years;ability to walk a flight of stairs (10 steps);ability to visit the study room independently.

The following exclusion criteria were applied:Subjects with acute medical contraindications (e.g., recent joint replacement surgery, dizziness) or symptoms that may cause regular attendance by another person for safety reasons (e.g., high risk of falls, palsies);Strong visual limitations that did not allow the test to be performed or the required interaction with the monitor without assistance;Inability to understand the study content and process it accordingly (e.g., due to cognitive or linguistic limitations).

All subjects fulfilling the inclusion criteria and expressing continuous interest in participating were invited to the study room where every person was informed about the study and its procedure and gave written informed consent to participate in the study. Afterward, an initial clinical assessment was done by a study nurse within the University of Oldenburg, covering the following herein relevant parameters, among others:Health status: body height, weight, BMI, medication review (assessed via the ATC scheme), and diseases such as stroke, diabetes, artificial joint replacements on the lower extremities, hypertension, and falls;Physical performance was conventionally assessed via stopwatch measures and scores:
-Stair Climb Power Test (SCPT) [30];-Timed “Up & Go” (TUG) [16];-Short Physical Performance Battery (SPPB) (includes 5SST) [31];-6 Minute Walk Test (6MWT) [32].

All participants were assessed by the same study nurse, measuring the performance of the TUG and 5SST via stopwatch measures. Since the study nurse previously conducted all considered assessments within the VERSA study (with around 250 participants), she was very routinized with the assessments of TUG and SST and was initially trained to handle and introduce the USS to participants. After the initial medical assessment, participants were introduced to handling the USS by the study nurse. They were informed to use the USS to conduct the TUG and 5SST on their own in an unsupervised manner.

To characterize the study sample, the DemTectindex, as assessed around six months in advance of this study, was considered. For the DemTect, the stages of age-corresponding cognitive performance (13–18), the stage of mild cognitive impairment (9–12 points), and indications for light dementia (<9 points) were differentiated.

The study was registered at the German Register for Clinical Trials (ID DRKS00015525) and was approved by the medical ethics committee of the University of Oldenburg (ethical vote: CvOUniversity Oldenburg medical ethics committee No. 2018-046) per the Declaration of Helsinki.

### 2.4. Postprocessing

Correlations between the manual supervised reference time measures and the unsupervised USS assessments (namely the LB- and IMU-based measures) were investigated based on the overall time of performance. To cover the medical relevance of the USS measures, both measures, the reference stopwatch measurements and the USS-based measurements, were collected in two independent test performances.

Initially, USS measures that fulfilled the termination criteria, discussed in Section 2.1, were excluded from further processing.

Results were evaluated via linear regression and Bland–Altman plots. For the linear regression analysis, the significance was assessed by the *p*-value, whereby a *p*-value <0.05 indicated a significant relation. The Pearson correlation coefficients were interpreted regarding this rule of thumb [33]: 1–0.9 meant a very high correlation, 0.7–0.9 a high correlation, 0.5–0.7 a moderate correlation, 0.3–0.5 a low correlation, and 0–0.3 a negligible correlation. Bland–Altman plots were also used to analyze the conformity between the overall test duration of both considered sensor systems.

For evaluations of the clinical validity, the results of the TUG and 5SST were categorized regarding the cut-off values of each test. Correspondingly, the categories for the TUG tests were <10 s (freely mobile), 10–19 s (mostly independent), 20–29 s (variable mobility), and >30 s (impaired mobility), per [16]. For the 5SST, a test duration of at least 10 s was confirmed as a sensitive two year predictor for developing disability in n = 4335 community-dwelling adults with a mean age of 72 years [18]. A cut-off time of 15 s (sensitivity 55%, specificity 65%) was confirmed as the optimal cut-off time for predicting recurrent fallers for 2735 subjects aged 65 and older in a good state of health [19]. We applied these cut-off values to both measures, the manual stopwatch measures, and the test-duration as identified by USS in post-processing and computed corresponding confusion matrices. For the cases with miscategorization, a manual visual inspection of the depth-image videos was conducted for reasoning about potential errors.

For data post-processing, MATLAB R2015a was used.

## 3. Results

### 3.1. Study Sample

Ninety-three participants were included in the study. According to the existing literature, sample sizes for the test-re-test reliability of IMUs provided a minimum of 24 persons to detect the TUG duration [34] and 41 persons for the 5SST duration [35]. Therefore, the reliability of the cohort size of 93 subjects was confirmed.

From the 93 initially included participants, one was excluded from initial screening due to a critical high Riva-Rocci blood-pressure measure [36]. One additional participant did not participate in the initial USS trial and had to be excluded, resulting in 91 subjects to be considered. The remaining 91 included participants’ age ranged from 73 to 89 years (average: 77.87, SD: 3.59). With 47 participants (52%) being female, the cohort held an age-representative gender distribution. With a DemTect index [37] (as assessed around six months before the given assessment) ranging from 6–18 with an average of 14.10 (SD: 2.9), the cohort generally resided still in a stage of age-corresponding cognitive performance (13–18 points). However, among the considered participants, 59 already had entered a stage of mild cognitive impairment (9–12 points), and three participants showed indications of light dementia (<9 points).

### 3.2. The Validity of Light Barrier-Based TUG Measurement

Evaluating the validity of the TUG, the overall TUG duration as measured by the USS light barriers (LBs) was compared with the corresponding stopwatch measures per participant (see Table 4 for a general overview of the results). As we showed in [28], the conducted stopwatch measures had low inter-rater variability and thus represented a suitable reference measure. However, since these two measures were conducted in different assessments (participants were expected to use the USS assessment without supervision), variations among the measures could be expected.

The results of a linear regression among the USS LBs and the stopwatch measurements are shown in Figure 5 and indicate a substantial association, resulting in the following equation:$$TUG_{Stopwatch}=TUG_{LB}*0.82+2.98$$

In addition, a significant correlation of r = 0.89 (*p* < 0.001) was identified via Pearson’s correlation coefficient. Furthermore, the Bland–Altman plot was used to analyze the conformity between both systems (see Figure 6). The mean difference of 1.49 ± 1.96 standard deviation of the difference (representing the 95% confidence interval) is also marked in the Bland–Altman plot at 3.09 and −0.10.

To evaluate the medical validity, we furthermore analyzed the classification regarding the cut-off values (<10 s, 10–19 s, 20–29 s, and >20 s) for the TUG. Therefore, we adjusted for the intercept and the correlation coefficient of the USS LB data as identified by the equation of the linear regression. Table 5 shows the categorization of the TUG results regarding the cut-off values for the TUG test [16]. Among the 12 participants deemed classified in the wrong category, four started the test too early, and eight delayed the start (as confirmed by visual inspection of the corresponding depth sensor video recordings).

### 3.3. The Validity of Inertial Measurement Unit-Based TUG Measurement

Among the IMU-based TUG measures, four measures had to be excluded due to technical or usage errors: next to two missing recordings, one participant paused the walking within the assessment, and one participant wore the belt upside down with the IMU in the opposite direction.

The results of a linear regression among the USS sensor belt and the stopwatch measurements are shown in Figure 7 and indicated a strong association, resulting in the following equation:$$TUG_{Ref}=TUG_{IMU}*0.71+2.36$$

In addition, a significant correlation with r = 0.78 (*p* < 0.001) was identified via the Pearson correlation coefficient. The Bland–Altman plot was used to analyze the conformity between both systems (see Figure 8). The mean difference of −0.65 ± 1.96 standard deviation of the difference is also marked in the Bland–Altman plot at 1.40 and −2.70.

To evaluate the medical validity of the IMU measures for TUG categorization, corresponding evaluations of the cut-off values (<10 s, 10–19 s, 20–29 s, and >20 s) were conducted. Therefore, we adjusted for the intercept and the correlation coefficient of the USS sensor belt data as identified by the equation of the linear regression.

Table 2 shows the categorization of the TUG results regarding the cut-off values for the TUG test [16]. Among the 16 participants incorrectly categorized, eight were categorized into a faster group and eight into a slower group.

### 3.4. The Validity of Light Barrier-Based SST Measurement

Evaluating the validity of the USS LB for the SST performance, the results of a linear regression among the USS LB and the stopwatch measurement are shown in Figure 9 and also confirmed a strong association, resulting in the following equation:$$SST_{Ref}=SST_{LB}*0.72+3.83$$

In addition, a significant correlation with r = 0.73 (*p* < 0.001) was identified via the Pearson correlation coefficient.

The corresponding Bland–Altman plot was used to analyze the conformity between both systems (see Figure 10). A mean difference of 0.29 was given. The mean difference ±1.96 standard deviation is also marked in the Bland–Altman plot at 5.28 s and −4.71 s. The confusion matrix regarding the applicability of the cut-off values in Table 6 and the corresponding medical validity indicated a low sensitivity degree for the LBs with 16 misclassifications (18%) for the 10s threshold and 14 misclassifications (15%) for the 15 s threshold.

### 3.5. The Validity of Inertial Measurement Unit-Based SST Measurement

Among the IMU-based 5SST measures, two assessment performances had to be excluded due to missing recordings. Six additional assessments were misclassified. Five participants conducted the assessment incorrectly (either by pausing in between or by conducting fewer 5SST cycles (three or four instead of the required five times). A total of 13 measurements were not considered.

The results of a linear regression among the USS sensor belt and the stopwatch measurement are shown in Figure 11 and confirmed as well a strong association, resulting in the following equation:$$SST_{Ref}=SST_{IMU}*0.97+0.35$$

In addition, a significant correlation with r = 0.87 (*p* < 0.001) was identified via the Pearson correlation coefficient.

The corresponding Bland–Altman plot was used to analyze the conformity between both systems (see Figure 12). A mean difference of −0.06 was given. The mean difference ± 1.96 standard deviation is also marked in the Bland–Altman plot at 3.32 and −3.43.

The confusion matrix of the cut-off values for the IMU-based measures in Table 6 indicated eight (5 + 3) miscategorizations (10%) for the 15 s threshold and seven (6 + 1) miscategorizations (9%) for the 10s threshold.

## 4. Discussion

The evaluation aimed to identify the validity of the USS’s time-based TUG and 5SST measures concerning stopwatch measures as a reference. The total test duration to perform the tests was validated separately for the USS LBs and the belt integrated IMU sensor in a supervised and unsupervised setting. Therefore, two timely distinct measurements were conducted: the first assessment via a conventional stopwatch measurement under the supervision of a study nurse and the second assessment via an unsupervised self-assessment by using the USS on a subsequent day (typically within one week after the supervised assessment).

Under all considered conditions, the USS measures showed a positive offset compared to the stopwatch measures (ranging from 0.35 s for the IMU-based SST to 3.83 s for the LB-based SST). This offset resulted partially from the higher sensitivity of the USS measures to detect the movement initiation earlier than the study nurse. Similarly, the end of the test might be measured longer. The minimal offset in the case of the IMU-based SST measure could be explained mainly by this effect because of holding a regression coefficient of 0.97. However, the offsets of the other three measures as well partially counterbalanced the regression coefficients (ranging from 0.71 to 0.82).

Considering the results of the correlation analysis via the Pearson correlation coefficient, this rule of thumb [33] was applied: 1–0.9 meant a very high correlation, 0.7–0.9 a high correlation, 0.5–0.7 a moderate correlation, 0.3–0.5 a low correlation, and 0–0.3 a negligible correlation.

For the TUG test, a high and significant correlation was shown by the regression analysis. With a correlation coefficient of 0.89 for the LB and 0.78 for the IMU measures, the validity of the LB was slightly stronger. Since recording the USS-based measures and the reference stopwatch measure took place in distinct trials, these results were especially promising and confirmed a low degree of inter-test variability.

Additionally, the conformity among the reference stopwatch measures and both sensors was evaluated via Bland–Altman plots (see Figure 6 and Figure 8). Since the mean difference was 1.49 for the LB and −0.65 for the IMU, both sensors held a fixed bias. The LB bias resulted from the LB not measuring initial reaction times. The findings agreed with a corresponding previous study [24]. The negative mean difference of the IMU might result from the sensor’s higher sensitivity to initial movements of the upper body and the delayed reaction times of the study nurse handling the stopwatch. Due to a reported Minimal Detectable Change (MDC) of the TUG test varying between 1.14 s [39] and 3.4 s [40], the given differences within the mean ± 1.96 SD might not be clinically relevant, and the USS sensor-based measures may be used in favor of stopwatch measures.

Comparing the number of detected TUG measures in Table 5, the IMU-based TUG detected four fewer test performances than the LB (one dropout compared to five). The additional missing performances resulted from these cases, as identified via manual visual inspection of the depth-image videos: besides two technical errors and one case of incorrect wearing of the sensor belt, one participant paused while walking during TUG testing. The last case indicated an invalid performance of the assessment, which could not be detected with the LBs. To detect cases of incorrect belt wearing and incorrect test execution, we will integrate a Bluetooth-connected IMU sensor to enable online processing of the movement patterns. Thereby, the USS will be able to identify such errors instantaneously and overcome such errors by presenting corresponding adjustment instructions.

Considering the miscategorization towards the cut-off values (as summarized in Table 5), the LB had 12 (13%) and the IMU had 16 (18%) miscategorization among the considered 91 participants compared to the stopwatch measures. The variation of measured test durations among the USS sensors and the reference stopwatch might partially result from the separate conduction of the tests for both measures and the associated intra-test fluctuations of the performance. Considering that another study’s LB measures and stopwatch measures were conducted in parallel for the TUG, no corresponding variations in test duration were found [24]. Considering this effect and the general low differences in total test durations among these miscategorized cases, we saw an indication of the sensors’ medical validity to detect the functional decline in the TUG test correctly.

For the 5SST, the correlation analysis also showed a significant and high correlation of 0.73 for the USS LB measurements and of 0.87 for the IMU measurements with the stopwatch measures.

The corresponding Bland–Altman plots indicated a strong conformity between both sensors and the stopwatch measure (see Figure 10 and Figure 12). With a low mean difference (LB of 0.29 and IMU of −0.06), both sensors had a slight intercept. The MDC of the SST differed in the literature depending on the observed study sample. For example, Goldberg et al. [41] found an MDC of 2.5 s in older females and Blackwood [42] an MDC of 3.54 s in older adults with early cognitive loss. Therefore, the recognized differences within the mean difference of ± 1.96 SD might not be clinically significant.

Table 6 lists the 5SST-based categorization for both sensors and indicates that for the IMU, 13 (14%) measurements had to be excluded. These 13 excluded measures contained six unintended misclassifications, two data losses, and five incorrect test performances that could not be detected by the LB. (In the beginning of the study, participants with a critical high body mass index were able to move the light barriers that were mounted at the seat unintentionally with their legs. These unintended movements of the LBs resulted in additional unintended LB activation and an earlier finishing of the 5SST. Within the study, this bug was handled by stabilizing the LB.). Since being only detectable via the IMU, those faulty performances were considered as valid measures for the LB.

The categorization based on the 5SST total duration failed for seven to 16 participants (as shown in Table 6), depending on the considered cut-off values and the sensor type. As for the TUG, the variations that caused these miscategorizations might relate to the separate performance of the USS and reference stopwatch measures and the potentially associated variations in performance, which was stated in the example of the five recognized faulty performances during the USS measure, which did not occur in the reference measure. Thereby, these variations might be at least partially a consequence of the natural fluctuation of the test performance. The differences in total test duration, causing such faulty transitions of SST categorization groups, were small. The validity of the USS sensors to measure the total performance duration of the SST was confirmed.

Thereby, the validity of both sensors to measure the time-based performance of the TUG and SST was confirmed.

Comparing the ICCof the USS to the related gait-speed and 5SST components of the SPPB kiosk [26] indicated that the USS achieved a comparable validity. The unsupervised TUG of the USS with an ICC of 0.89 (<0.001) and 0.78 (<0.001) achieved comparable validity to the supervised gait speed assessment of the SPPB kiosk with an ICC 0.84 (<0.001). For the SST, the SPPB kiosk achieved a better ICC of 0.99 (*p* < 0.001) compared to an ICC of 0.87 and 0.73 (*p* < 0.001) for the USS. The lower ICC in the case of the USS might result from the comparison of two separate test executions, while for the SPPB kiosk evaluation, the same test execution was considered for both conditions. This overall similar validity between supervised and unsupervised technology-supported assessments indicated the applicability of the proposed USS for unsupervised assessments.

With the reported results being evaluated with relatively healthy older adults, the current USS might not be applicable for users with significant motor or cognitive limitations. To apply the USS for community-dwelling older adults, some adjustments should be considered. A useful approach would be an initial examination by general practitioners to exclude persons in case of safety concerns (e.g., a high risk of falls, orthostatic issues). Additionally, adjustable/pop-up railings might be a suitable added safety feature for some users. For users with cognitive limitations, navigating the USS interface might be more challenging and may require personal assistance. However, this loss of control would be taken into account as a relevant indicator for cognitive decline to be investigated aside from corresponding motor abnormalities [43].

Along with the design phase and the validity study of the USS, a user experience study was conducted to identify user expectations, challenges, and experiences. Corresponding findings regarding the user experience and identified necessary improvements will be addressed in an upcoming article.

## 5. Conclusions

The article introduced the Unsupervised Screening System (USS), a screening system for unsupervised assessment of older adults by the TUG and SST for the early detection of functional decline. Its initial evaluation of the validity for measuring the total test-duration of the TUG and SST assessment, as measured via (1) the Light Barriers (LBs) and (2) an inertial sensor integrated into a sensor belt (IMU), was conducted in a trial with 91 older adults and via reference stopwatch measures. The results indicated both considered sensors and assessments as having high and significant correlations. By applying established cut-off values for the detection of functional decline (in mobility impairment for the TUG and degree of muscle power for the SST), the USS’s medical sensitivity could be confirmed.

Comparing the measures of the LB and the IMU sensor, we found that the additional context awareness of the IMU sensor enabled the detection of faulty test performances. Thereby, the results indicated that while the timing information of the LBs was slightly stronger, additional sensors such as the IMU were required to detect performance errors. Consequently, we foresee increased reliability resulting from data fusion of multiple sensor types to monitor correct performance by users and to extract additional quality performance indicators. We are looking forward to evaluating the system’s validity for detecting functional decline based on the progression of frequently (e.g., monthly) conducted measures.

## Figures and Tables

**Figure 1 sensors-20-02824-f001:**
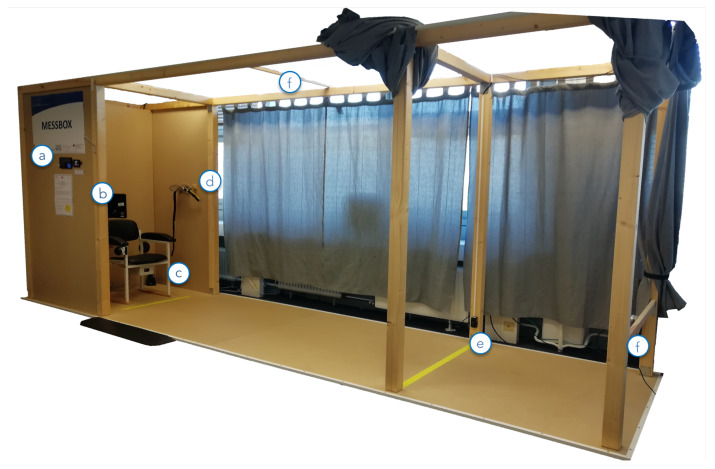
The USS consisting of: (**a**) the introductory display and an RFID authentication device, (**b**) the main display for user-interaction, (**c**) the integrated aTUG chair with additional LBs, (**d**) a sensor-belt including an inertial sensor, (**e**) light barriers at a 3 m walking distance, and (**f**) the Intel RealSense™ D435 Depth image camera placed approximately 4 m from the chair and from the upper level facing it.

**Figure 2 sensors-20-02824-f002:**
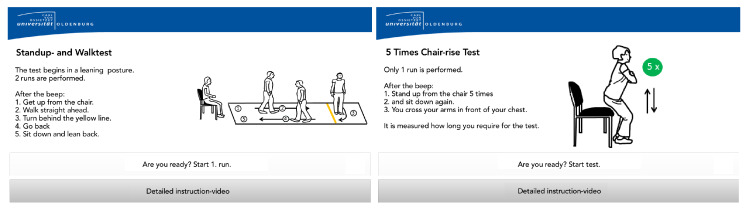
Examples for the instructional screens for the TUG and 5SST. Since the prototype was only evaluated and implemented with German native speakers, the screens were translated into English for this article.

**Figure 3 sensors-20-02824-f003:**
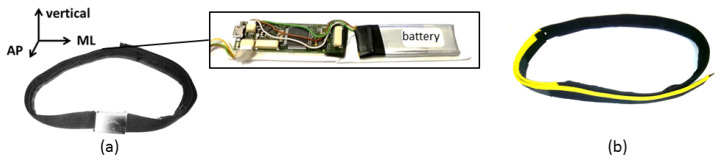
The applied sensor belt (**a**) in its original design with the integrated triaxial inertial sensor system and its orientation and (**b**) in its adapted form. Abbreviations: ML = mediolateral and AP = anterior-posterior.

**Figure 4 sensors-20-02824-f004:**
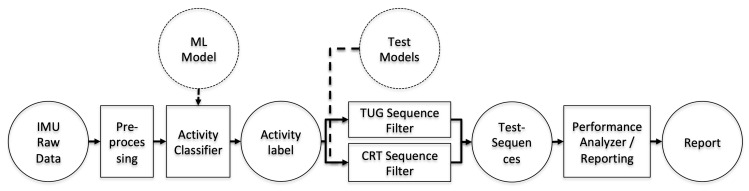
System workflow for processing raw IMU data and reporting via machine-learning-based activity classifiers, a rule-based model for test detection, and performance analyses.

**Figure 5 sensors-20-02824-f005:**
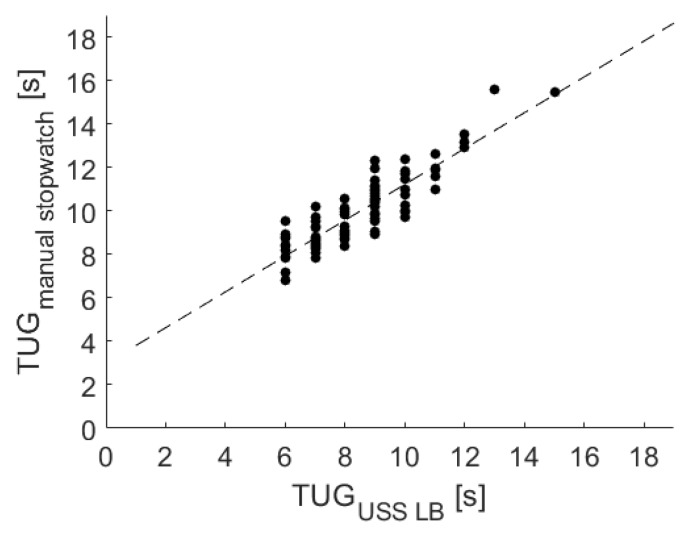
Linear regression among the manual stopwatch and the USS’s LB measurements for the TUG test.

**Figure 6 sensors-20-02824-f006:**
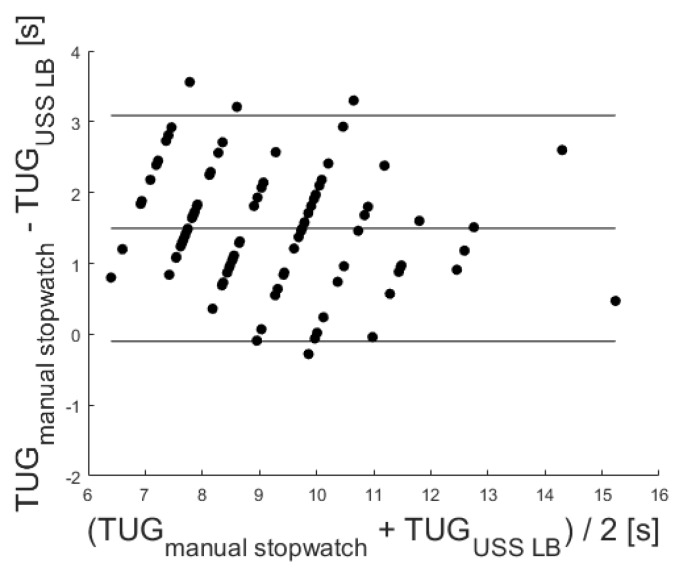
Corresponding Bland–Altman plot. Included levels indicate mean Diff ± 1.96 SD (−0.10 to 3.09 s).

**Figure 7 sensors-20-02824-f007:**
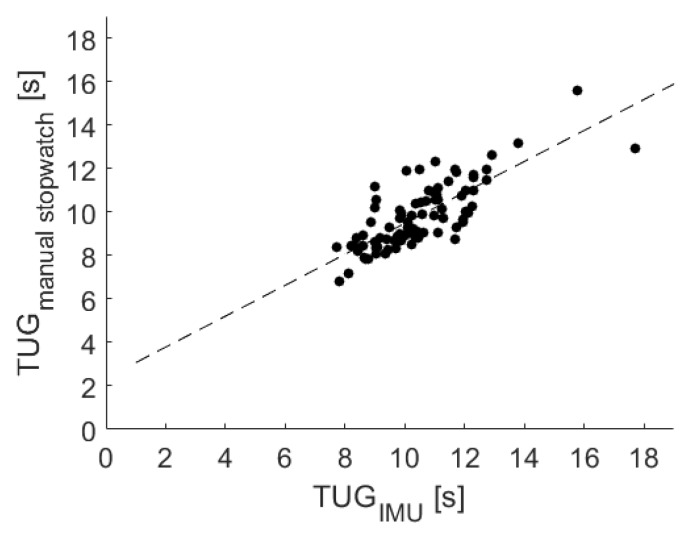
Linear regression among the manual stopwatch and the USS’s sensor belt measurements for the TUG test.

**Figure 8 sensors-20-02824-f008:**
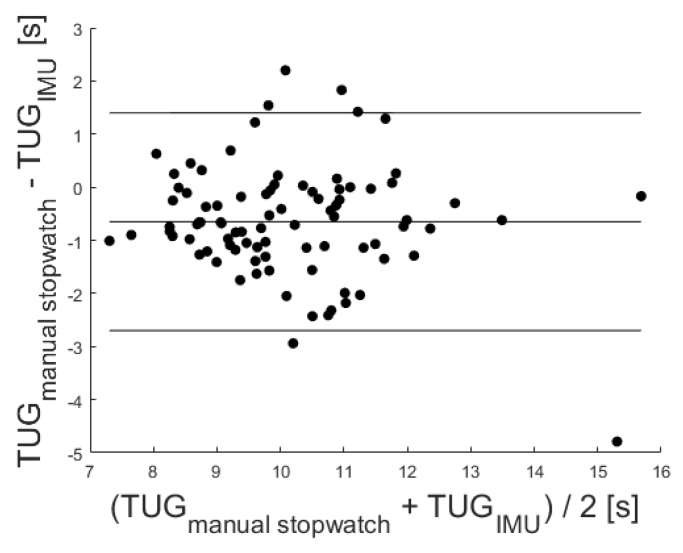
Corresponding Bland–Altman plot. Included levels indicate mean Diff ± 1.96 SD (−2.70 to 1.40 s).

**Figure 9 sensors-20-02824-f009:**
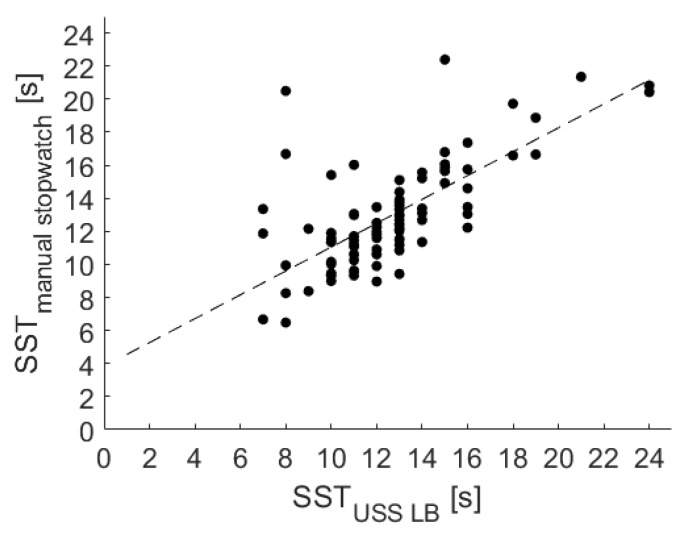
Linear regression among manual stopwatch measurements and the USS LB measurements for the SST.

**Figure 10 sensors-20-02824-f010:**
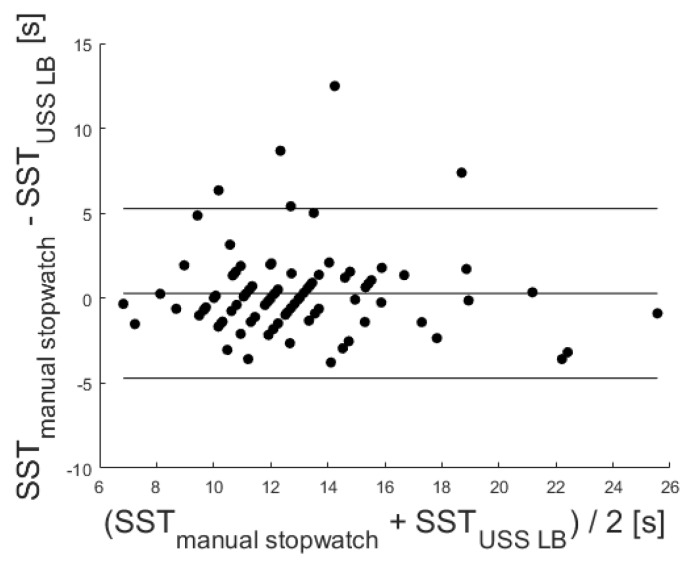
Corresponding Bland–Altman plot. Included levels indicate mean Diff ± 1.96 SD (−4.71 to 5.28 s).

**Figure 11 sensors-20-02824-f011:**
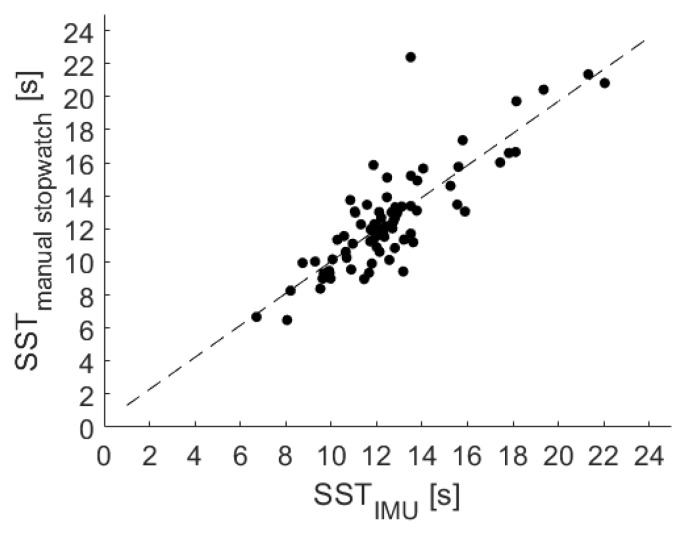
Linear regression among manual stopwatch measurements and the USS sensor belt (IMU) measurements for the SST.

**Figure 12 sensors-20-02824-f012:**
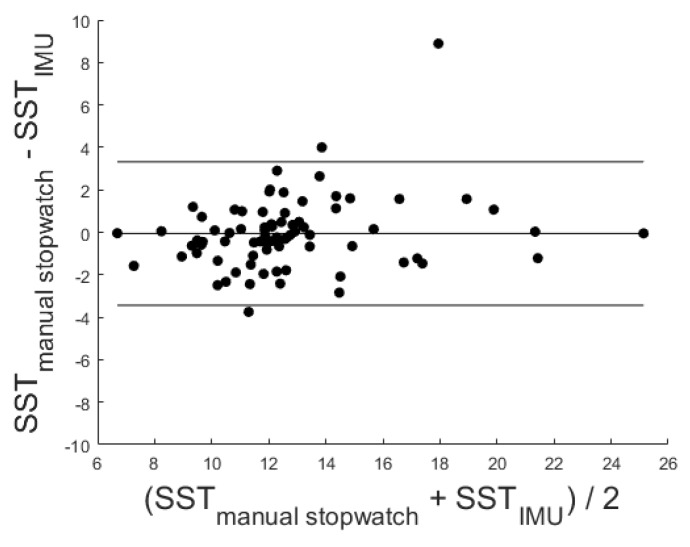
Corresponding Bland–Altman plot. Included levels indicate mean Diff ± 1.96 SD (−3.43 to 3.32 s).

**Table 1 sensors-20-02824-t001:** The specifications of the sensors included in the belt.

Sensor	Frequency	Resolution	Measuring Range
Accelerometer (Bosch BMA180)	100 Hz	12 bit	±16 g
Gyroscope (Bosch)	100 Hz	14 bit	±2000 $\deg\cdot s^{-1}$
Magnetometer (MEMSIC MMC314xMR)	100 Hz	12 bit	±500 μT
Barometer (Bosch BMP085)	100 Hz	14 bit	-

**Table 2 sensors-20-02824-t002:** *F*1-scores of the classification methods for the different classifiers: Boosted Decision Trees (BDT) and Multilayer Perceptrons (MLP). Based on [27].

Classifier	*F*1-Score (%)	Method
State	96.6	BDT
Static	97.3	MLP
Dynamic	97.5	MLP
Transition	94.8	MLP

**Table 3 sensors-20-02824-t003:** Included test classification models for the following listed activities: sit-to-stand (↑), Walk (W), Turn (T), Stand (S), sit (P), stand-to-sit (↓); [] indicates optional activities, which might change per performance.

Test	Model
TUG	↑WTWT↓
	↑↓WTWT↓
	↑↓TWT↓
	↑↓T↓
	↑↓WT↓
	↑TWT↓
	↑↓WTWTW↓
	↑WTWTW↓
	↑WT↓
SST	5x:↑[S]↓[P]

**Table 4 sensors-20-02824-t004:** Characteristics of the measures in seconds. Overall, 91 participants were considered. LB: Light Barriers, IQR: Interquartile Range, Min: Minimum, Max: Maximum, TUG: Timed “Up & Go” Test, 5xSST: Five Times Sit-to-Stand Test.

Sensor	Parameter	Min	1 IQR	Median	3 IQR	Max
TUG stopwatch		7	9	10	11	16
LB	TUG USS	6	7	8	9	15
	TUG difference	−0.28	0.945	1.47	1.95	3.56
Sensor belt	TUG USS	7.73	9.27	10.24	11.265	17.7
	TUG difference	−4.79	−1.14	−0.68	−0.075	2.2
SST stopwatch		6	11	12	15	25
LB	SST USS	7	10.5	12	14	26
	SST difference	−3.78	−1.01	−0.03	1.135	12.49
Sensor belt	SST USS	7.73	11.045	12.245	13.5	25.16
	SST difference	−3.75	−1.07	−0.15	0.67	8.89

**Table 5 sensors-20-02824-t005:** Comparison of the categorization of the cut-off values for the TUG test [16]. All measurement values were adjusted via the corresponding linear regressions and rounded. Since no TUG performance took over 19 s, the additional classes were excluded.

		Stopwatch	
	Points	≤10 s	10–19 s
LB	≤10 s	45	4
	10 –19 s	8	34
IMU	≤10 s	45	8
	10–19 s	8	26

**Table 6 sensors-20-02824-t006:** Comparison of the categorization of muscle power in the SST regarding the cut-off values of [38].

		Stopwatch			
	Points	<15	≥15	<10	≥10
LB	<15	65	4	0	0
	≥15	10	12	0	0
	<10	0	0	4	4
	≥10	0	0	12	71
IMU	<15	60	3	0	0
	≥15	5	10	0	0
	<10	0	0	9	1
	≥10	0	0	6	62

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
