# Peer review of "Measurement System for Unsupervised Standardized Assessment of Timed “Up & Go” and Five Times Sit to Stand Test in the Community—A Validity Study"

_sensors, 2020, doi:10.3390/s20102824_

Round 1

Reviewer 1 Report

I commend the authors on a well designed and implemented study of the USS. You have identified a gap that is certainly a concern in regards to routine assessment in older adults. You have used a large sample of older adults that included some adults with mild cognitive decline which is essential for this age group. 

Below are some suggestions/comments.

Line 43: should be seems, not seams

Instructions for the 5xSST and TUG – I would change “run” to trial/test or something similar in case participants somehow take the term run literally (particularly for the TUG). The instructions for the TUG should also say "at a comfortable walking pace, or your normal walking speed".

Line 121: I wonder if a 5 second limit might be more appropriate as some older adults can take quite a while to initiate movement following an auditory cue. 

Line 125: 30 seconds may be too short a cut-off for some older adults (although those that require >30seconds might also require close supervision). I see from results that no-one took this long but in a wider community setting this might be required

Line 178: please provide more information (or specific examples) regarding potential limitations to provide a better sense of which individuals could safely perform the tests

Line 260, 279, 294: why not use 95% confidence interval here to identify variability

Discussion: can you please refer to the possible limitations of the system in regards to the population that can be tested and potential safety considerations (ie. potential for a fall during testing)? Have you considered including adjustable/pop-up railings as an added safety feature? I realise that you would expect people doing the test unsupervised to have a certain level of independence (ie. able to climb a flight of stairs). The physical abilities of your sample are clear given the inclusion criteria, but can you please expand on the potential use/applicability of the system for community dwelling older adults. You have discussed the accuracy of the system but it's real world application needs to be addressed further, accepting that future feasibility studies are likely to occur. 

Reviewer 2 Report

General comments: the authors have published several papers related that are listed as references, however as a general comment it is not always clear what was done before and what adds this article.

Minor comments: ref 11 is not complete

Introduction:

When introducing related works as aTUG, is that not unsupervised? It seems so according to the text, so what is adding the new version? Authors state that “is still needed” in line 62, what is needed? There are indeed unsupervised methods for functional measurements, the novelty is in the combination of TUG and 5SST¿?

Method:

As explained in the introduction, authors use as gold standard measurement the comparison with manual stopwatch. In all 91 persons, the professional doing the manual measurement is the same person? We already know there are high between velocity in pressing the buttons amongst professionals, ensuring that the stopwatch measure is not varying is important. More information about how this was handle is needed.

The system seems complicated, more information about where is suitable is needed, for instance in the discussion, where is not included. Also, considering the potential user is an old and maybe frail person, how the authentication works concerns me in terms of security and usability. Authors need to comment on whether they found or not problems with it. In general, information about usability of user experience during the evaluation is not included. Some discussion about how is so, is needed. Do they feel comfortable with the ambient system and the wearable one? How are they introduced into the tests? Who explains to them the process?

From line 137: this paragraph refers to another paper written by the authors, however, some information about it, as data used (i.e from how many different adults) is missing. Are they the same? Are they different persons than the 91? In that case, how the authors assess validity in the sense that both samples are comparable? Here also reflects my general comment about the previous work already published. It may help a lot the reader to state clearly from the beginning and of course in methods, which are exactly the contributions of previous and more importantly current paper.

Results are very descriptive.

Discussion:

12 and 16 participants out of 91 seems a high number and more information about how the system detects, for instance “starting too early”, and the reaction, is the system then asking the user to walk again? Error handling needs more explanation. (this comment is related also to line 351 and the following)

Results of current paper are compared with authors’ previous work, but comparison with others is missing.

Information on how authors plan to improve technical errors handling, for instance due to bad-placement, or rapid or late start is missing.

Conclusion:

Line 394: authors do not transmit confidence when they only say “could” be used being medically accurate.

I miss along with the technical results, some insight about real users actually using it, their experiences and feelings, this is crucial to make sure such a system can improve managing functional decline and frailty.

References: the proportion of self-references is very high, and some comparison with other related work on how to measure functional variables outside clinical settings can be improved.

Round 2

Reviewer 2 Report

The article has significantly improved. Authors have answer to all questions introducing additional information where requested.

My concern is about the user’s experiences descriptions included in the discussion. It seems odd to discuss concrete experiences when they are not included in the results. My suggestion is just to mention that the user experience is important to be studied and considered, but it is not part of the technical validity that is the aim of the paper. In this regard, maybe it better to state from the very beginning that the aim of the paper is to present the technical validity, the feasibility, and then present as future or further work the assessment of the user experience and the improvements needed in that sense. Thus, it implies removing most information between lines 421-433.
